# Measuring the Supply Chain Performance of the Floricultural Sector Using the SCOR Model and a Multicriteria Decision-Making Method

Luís Oswaldo Rodríguez Mañay [1], Inmaculada Guaita-Pradas [2] and Inmaculada Marques-Perez [2,*]

[1] Facultad de Ciencias Administrativas, Universidad Central del Ecuador, Quito 170129, Ecuador; lorodriguez@uce.edu.ec

[2] Faculty of Business Administration and Management, Economics and Social Sciences Department, Universitat Politècnica de València, 46022 Valencia, Spain; iguaita@esp.upv.es

* Correspondence: imarques@esp.upv.es

**Abstract:** This study aims to highlight the usefulness of studying the performance of supply chains (SC) at the sectoral level in greater detail through the combination of a disaggregated supply chain operations reference (SCOR) model, with a multicriteria decision-making approach, specifically using an AHP, to adjust the analysis to the particularities of the sector under study by stakeholders' judgements. The methodology was applied to the Ecuadorian flower industry, and the data for the analysis was from a survey of a group of companies that represent this sector. In addition, a focus group of SC experts weighted the model constructs as part of the analytic hierarchy process (AHP), and then the performance level for each construct was determined. According to the results methodologies, this model allows the classification of companies by their performance, as well as the performance of the aggregate sector. The processes that Ecuadorian flower companies need to improve on are planning, procurement, and manufacturing. The study's main contribution is developing a general framework for measuring the overall performance of SCs and how the results are obtained. This tool could help managers, consultants, industries, and governments to assess the performance of SCs, as well as improving SC management in order to increase the sector's competitiveness in the international market.

**Keywords:** supply chain performance; floricultural sector; SCOR; AHP

## 1. Introduction

Supply chains (SC), which are understood as a system of people, organizations, activities, resources, and data that are involved in the flow of products or services from the supplier to the customer [1] have developed continuously over the past forty years [2], especially during the months following the outbreak of the pandemic [3]. SCs evolve for two reasons: (i) to improve their performance and the system's functioning, as well as the elements that make it up, and (ii) to ensure consumer satisfaction [4]. Recently, the COVID-19 pandemic has exposed the vulnerability of supply chain risk management [5]. The concept of *supply chain management* (SCM) was first introduced in the 1980s to express the need to integrate all the processes of a supply chain, from the end-user to the original suppliers [6,7]. Since then, plenty of research has been undertaken both to study supply chain management in various fields of activity (industry, transportation, distribution, and agriculture, among others), as well as to measure and determine the ability of different SC processes to achieve their set goals, and also, to identify processes which could be improved in order to make SCs more effective and efficient. Among the most recent works, the following are worth highlighting: supply chain risk management (SCRM) [8], environmental supply chain management (GSCM) [2], and supply chain performance management (SCMP) [9,10], as well as those works exploring the use of technologies, such as artificial

intelligence (AI) [11,12], the Internet of Things (IoT) [12], 3D printing [12], big data [12], and blockchain [12,13].

It should also be stressed that, nowadays, supply chain management (SCM) needs to adapt to more dynamic environments characterized by competition, rapidly evolving technologies, and higher consumer expectations for responsiveness [14]. All of these circumstances put pressure on supply chains to be more integrative and collaborative [4]. SC integration enables SC systems to shorten their response time because it allows the frequent and rapid changes in markets and demand to be managed [2]. Silvestro and Lustrato [15] emphasized the importance of integrating physical supply chain activities for several reasons: (1) it provides quick responses to fast-moving markets under conditions of demand uncertainty [16]; (2) it enables a closer collaboration between buyers and sellers along the supply chain, resulting in significant reductions in delivery times and costs [17,18]; (3) an integrated supply chain works better than each process on its own [19,20]; and (4) it maximizes information visibility through the use of the Internet and the involvement of all the parties in the supply chain [21,22]. Given that information transparency along the supply chain has become a priority for buyers and suppliers, highly complex supply chain networks tend to improve their performance when integrated [13,23].

There are several models and techniques for assessing SC performance, of which the following stand out: (a) the supply chain operation reference (SCOR) model, which is a model that describes, communicates, assesses, and identifies opportunities for improving workflow efficiency [4]; (b) the Global Supply Chain Forum (GSCF) model, which provides a systematic overview of the balance, alignment, and management of SC technological capabilities to achieve successful management [4]; (c) the Triple-E model, developed by Simao et al. (2021) [10], which focuses on three performance dimensions: efficiency, efficacy, and environmental impact [10]; and (d) the BSC model, developed by Kaplan and Norton, which allows managers to obtain an overall view of a supply chain's performance [24,25].

Developed and endorsed by the Supply Chain Council (SCC, focuses primarily on defining the core processes that make up a supply chain system) as an industry-standard diagnostic tool, the SCOR model emerged in 1996 and, since then, it has evolved from its initial design to its current 12th version (The SCC with American Production and Inventory Control Society (APICS) produced the latest SCOR version, 12.0, in 2017) [10]. It is a powerful tool for structuring, assessing, and comparing supply chain practices and performance [26,27]. Furthermore, it is known to be an integrated approach based on the idea that the SC is an interconnected structure that combines SC processes, performance metrics, best practices, and technology into a single framework for the effective communication and the continuous improvement of the SC [5]. Moreover, it has been increasingly used by practitioners and academics involved in value chain management [28] and, in general, it is a global benchmark that enables the comparisons of SCs [29].

In recent years, several studies of supply chain management have combined the SCOR model with multi-criteria techniques to improve the analysis of SCs. Table 1 provides a list of these combinations, along with the works.

**Table 1.** Combinations of the SCOR model with multi-criteria techniques for studying supply chain management.

| Authors | Methods Applied | Aim |
|---|---|---|
| Nisa Afifa & Santoso, 2018 [30] | SCOR–FUZZY–ANP | Supply chain risk management |
| Effendi et al., 2019 [31] | SCOR–DEMATEL | Assess the performance of green supply chain management |
| Büyüksaatçi Kiriş et al., 2019 [32] | SCOR–FUZZY DEMATEL | Evaluate suppliers' performance |

**Table 1.** *Cont.*

| Authors | Methods Applied | Aim |
|---|---|---|
| Wang et al., 2018 [33] | SCOR–AHP–TOPSIS | Select suppliers |
| Lima-Junior & Carpinetti, 2016 [34] | SCOR–FUZZY TOPSIS | Assess suppliers |
| Lhassan et al., 2018 [35] | SCOR–BPMN | Map supply chain processes |
| Teixeira & Borsato, 2019 [36] | SCOR–BPMN | Dynamic formation of supplier networks to optimize SCs |
| Liu et al., 2018 [37] | BSC–SCOR | Green supply chain management |
| Wang, Yang, et al., 2019 [38] | SCOR–FANP–TOPSIS | Select suppliers |
| Wang, Van Thanh, et al., 2019 [39] | SCOR–FANP–VIKOR | Select suppliers |
| Wang, Tsai, et al., 2020 [40] | SCOR–AHP–DEA | Select suppliers |

Source: Authors' review.

In this regard, various studies have combined the analytic hierarchy process (AHP) [41] with the SCOR model in supply chain analyses [42]. The most relevant works are listed in Table 2.

**Table 2.** Research using both the SCOR model and AHP approaches to improve supply chain performance.

| Authors | Techniques | Aim |
|---|---|---|
| Kocaoğlu et al., 2013 [42] | SCOR–AHP–TOPSIS | Decision-making process in a manufacturing company for the construction industry |
| Wang, Hoang Viet, et al., 2020 [43] | SCOR–ANP–FAHP–PROMETHEE II | Select suppliers in textile industry |
| Bukhori et al., 2015 [44] | SCOR–AHP–Cause Effect Diagram | Identify performance issues in poultry supply chain by a poultry company |
| Palma-Mendoza & Neailey, 2015 [45] | SCOR–AHP–BPR | Redesign business processes in an Airline MRO supply chain |
| Sellitto et al., 2015 [46] | SCOR–AHP | Measure SC performance in the Brazilian footwear industry |
| Sutoni et al., 2021 [47] | SCOR–AHP | P.T. performance X for the production, warehouse, and shipping of goods in a company |
| Nguyen et al., 2021 [48] | SCOR–AHP | Measure performance of the Vietnamese coffee supply chain |
| Defrizal et al., 2020 [49] | SCOR–AHP | Analyze how rice supply and rice supply chain systems work |
| Novar et al., 2018 [50] | SCOR–AHP | Monitor the metrics of a supply chain measurement system |

Source: Authors' review.

The SCOR model is based on a hierarchical structure with four different levels. Level 1 presents the different types of processes and identifies the scope and content of the supply chain. Level 2 presents the process categories that include the operations (sub-processes), while Level 3 corresponds to the process elements that form the individual process configurations (tasks that are grouped by activities in each sub-process) [48]. The first point to consider, when analyzing the SCOR model processes, is to check which ones need to be analyzed, as well as the level of disaggregation, i.e., whether they are primary processes, sub-processes, specific activities, or tasks. In addition, it is necessary to establish a measur-

ing system with which the values that reflect the level of performance of these processes can be calculated [51].

In general, it can be observed that previous SC assessments using the SCOR model and the associated performance metrics predominantly analyzed supply chains' main processes, but very few of them considered a division into sub-processes and activities, and almost none on them considered a disaggregation into tasks [52,53]. However, an analysis of the individual processes, sub-process, activities, and tasks could help to better identify where the problems originate in each process; in other words, it would enable us to identify which process, or sub-processes, activities, or tasks are more critical, why they are critical, what the causes are, and how they can be corrected.

This approach has been applied to the Ecuadorian flower industry.

In distributing and selling perishable products, such as flowers, supply chain management is a crucial and decisive element in improving their efficiency, productivity, and the overall distribution and sale processes. Ecuador is the third-largest producer of cut flowers in the world, where flower companies are a significant source of income and employment for this country [1]. The Ecuadorian floriculture industry is characterized by short product life cycles, a wide product variety, volatile and changing demand, and long and inflexible delivery processes [2,3]. Since 2021, due to the COVID-19 pandemic, it has also been beset by international trade and transport problems [4], which have affected the production and marketing of thousands of products traded around the world. With regard to the Ecuadorian flower sector, in particular, the greatest impact of the COVID-19 crisis has been due to a rise in the price of inputs and fertilizers [5], as well as the lack of air freight companies that could deliver floral products on time, with the required quality [5,6]. These constraints and difficulties are currently exposing the supply chain (SC) management to a variety of risks and uncertainties [7,8]. Any attempt to improve the distribution channels in the floriculture sector requires a detailed analysis of its supply chain performance. The proposed performance analysis model was applied to a set of flower companies to assess how well the supply chain was performing at the individual level, and to identify the problems. The individual values were then aggregated to establish whether the supply chain was working well in sectoral terms, and similarly, where the problems lay. Currently, the Ecuadorian flower sector does not have a methodology or model to measure the performance of the supply chain. We apply this proposal to the Ecuadorian flower industry.

The content of the manuscript is structured as follows. First, the SCOR model approach, followed by the analysis of the floriculture supply chain, is explained. Then, consultations that are carried out with the sector's companies to obtain each company's performance data is described, as well as the order of processing and aggregating the survey results to work out the individual performance values. Next, using an AHP, the performance results are interpreted and discussed by analyzing the sector's performance through the individual and aggregated results. Finally, the practical and theoretical implications of the proposed methodology are discussed, as well as the most relevant issues and suggestions for future research.

With this purpose, here, we present a methodology for examining supply chains' levels of performance at the sectoral level, combining the SCOR model, that is disaggregated to Level 4, with a multi-criteria methodology (AHP) to adjust the analysis to the specificities of the sector under study, based on stakeholders' assessments. In particular, by applying the proposed methodology, we can determine which processes are the most critical, and why, as well as the causes of performance problems and how those can be corrected.

## 2. Materials and Methods

Figure 1 summarizes the methodology used to analyze the Ecuadorian flower sector based on the structuring of a supply chain, as defined by the SCOR model, in combination with an AHP approach.

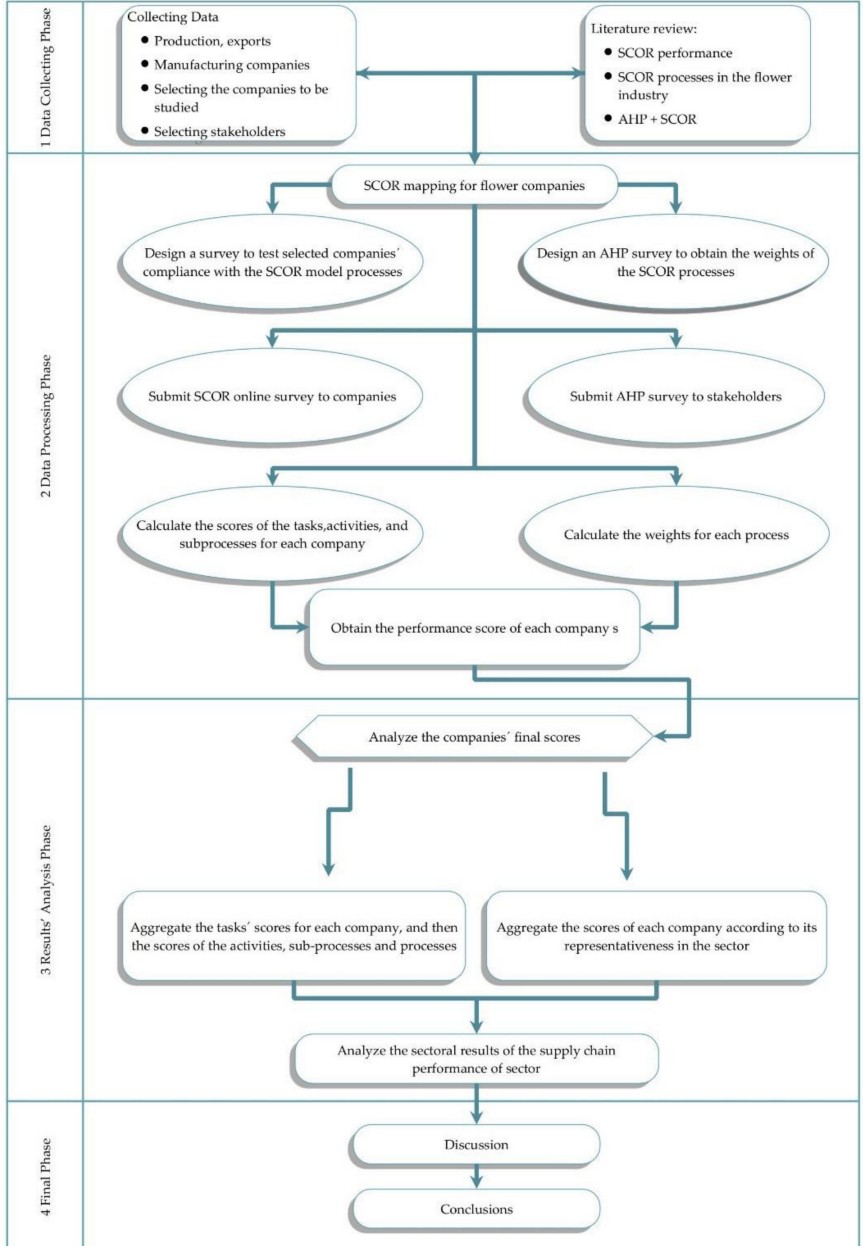

**Figure 1.** Methodology. Source: Authors' diagram.

As mentioned earlier, the 12th version of the SCOR model establishes a performance-analysis system with up to four levels. Thus, in addition to the first level of the supply chain, which is composed of six main processes (planning, procurement, manufacturing, distribution, return, and management), three more levels can be differentiated, namely, sub-processes, activities, and tasks, where each one might influence the main processes' performance and should, therefore, be analyzed [54].

Previous studies dealing with SC measurement, using the SCOR model, examined four, five, or six of its processes. In this study, we examined the planning, procurement, manufacturing, distribution, and return processes, which are those that are directly linked to the supply chain [55].

For example, process 1, planning, is broken down into three sub-processes [56] (see Figure 2). Each of these sub-processes is, in turn, disaggregated into different activities. For example, sub-process 1.1, supply chain planning, is decomposed into four activities, each of which is then divided into tasks (see Figure 2).

As previously pointed out, the greater the disaggregation, the better the analysis can identify the failures and where action is needed [57,58]. Our methodological proposal is to disaggregate each of the five main supply chain processes, up to level 4, which corresponds to the individual tasks.

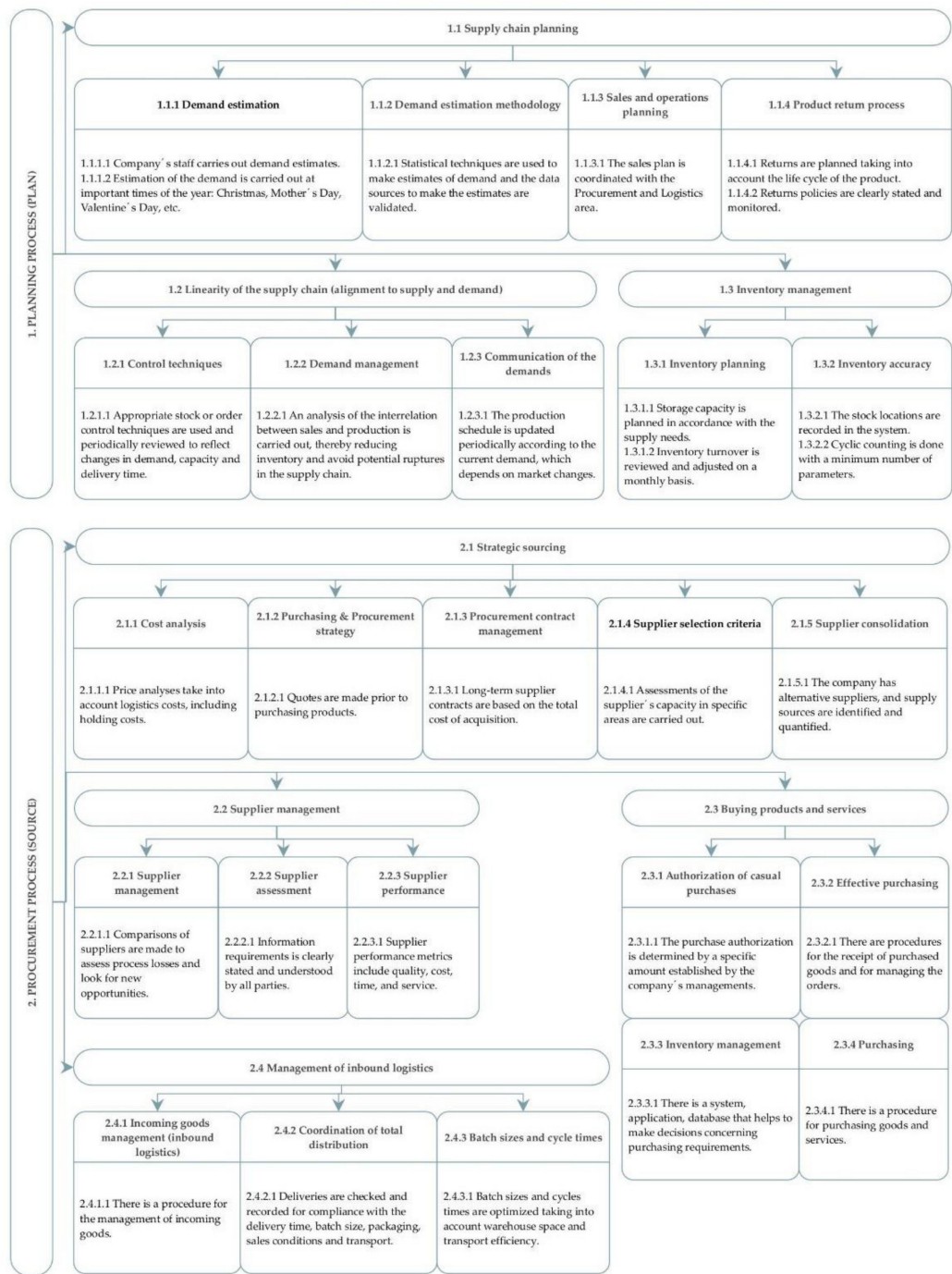

**Figure 2.** *Cont.*

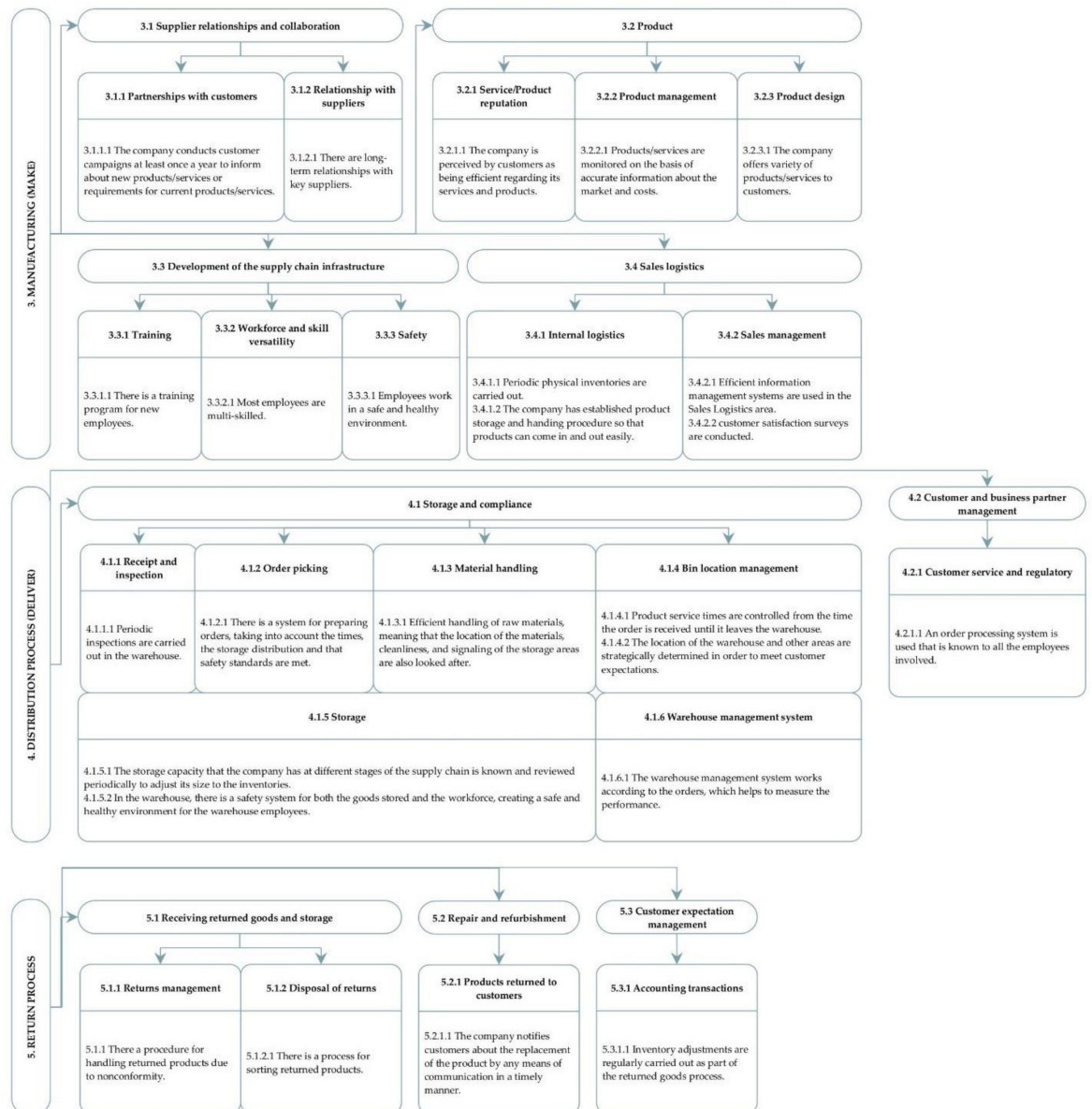

**Figure 2.** Disaggregation of the SCOR model's supply chain main processes into sub-processes, activities, and tasks. Source: Authors' diagram.

The proposed evaluation method is to assess all the processes and activities of the SC, in regard to their compliance with the standards. Thus, our SC performance assessment is based on checking whether the tasks, activities, sub-processes, and processes were completed or not [51]. Consequently, each company was sent a survey with a proposed breakdown of the SC and was asked to indicate whether or not it carried out the different individual tasks. A good performance involved completing each one of the defined tasks (i.e., the companies replied to dichotomous questions with yes or no answers), which meant that all activities, sub-processes, and processes were performed. After collecting the answers, we assigned a one to those tasks that were performed (for answering YES), and for those that were not carried out, a zero was assigned (for answering NO). Additionally, it is necessary to first carry out a systematic evaluation of each particular process and establish how the results are aggregated afterward to obtain a metric for measuring the SC's level of performance within the sector. To calculate the SC's overall performance, the aggregate value of the performance index must be calculated by weighting each SC process according to its relevance in the sector of activity. By using the AHP technique, it

is possible to distinguish the importance of each process of the SC when aggregating the data. This distinction is made by the sector's stakeholders, based on the importance they attach to every process or activity, since the aim is to provide a metric that considers the particularities of each sector. To aggregate the single values obtained from the rating of the tasks, activities, sub-processes, and processes, Aliaga Rota et al. [51] proposed using the average of the separate scores given for each sub-process, which, in turn, are gathered from the average of the scores obtained by the activities involved in that sub-process, and so on [51,59]. The aggregation of the results is carried out, considering the importance given to each process by the stakeholders participating in the AHP [60]. As a result of this aggregation, the SC performance of each company in the sector can be analyzed.

The AHP is a technique by which experts in a given field make pairwise comparisons in order to derive priority scales. Furthermore, it provides an algorithm to solve complex decision-making problems that are broken down into a hierarchy [61,62]. This method involves two main steps [63]. First, each stakeholder completes a pairwise comparison survey, which is designed based on the hierarchy previously established, indicating which of the two elements that are compared they consider to be more important and, using Saaty's scale (Table 3), how much more important they are.

**Table 3.** Saaty's scale.

| Intensity Scale of Importance | Definition |
| --- | --- |
| 1 | Equal importance |
| 3 | Moderate importance |
| 5 | Strong importance |
| 7 | Very strong importance |
| 9 | Extreme importance |

Source: Leal [63].

The second stage of the AHP is to calculate the vector of priorities, according to the following formula:

$$p_{rj} = \frac{1}{a_{ij} * \sum_{k=1}^{n} \frac{1}{a_{ik}}} \tag{1}$$

where $j$ is the element for which the priority is calculated, $i$ is the base element for the comparison, $a_{ij}$ is the value of the alternative $i$ that is compared with the alternative $j$, by the criteri $k$, $a_{ik}$ is the value of alternative $i$ for the criteria $k$, $p_{rj}$ is the priority of the alternative $j$ against the considered criterion, and $n$ is the number of criteria.

The coherence of the preferences of stakeholders was studied based on the "consistency", which should be taken into account in order to consider whether opinions are valid for determining the priorities. The consistency analysis requires calculating the "consistency index" (*CI*) of Saaty's Scale for each preferences matrix.

$$CI = \frac{\lambda_{max} - n}{n - 1} \tag{2}$$

The "consistency ratio" (*CR*) is calculated from the *CI*. The *CR* is a ratio of the *CI* and *RI*:

$$CR = \frac{CI}{RI} \tag{3}$$

where the *RI* is the average value of the *CI* of pair-wise comparisons matrices of the same order, randomly obtained. When the *CR* is less than 10% (0.1), the matrix is considered to offer acceptable consistency. Saaty's scale calculated the random indices of the RI for different matrix sizes to obtain *CR*.

There are two possibilities, when aggregating results, to analyze the company performance at the sectoral level. The first one is to aggregate the individual results obtained from individual analyses. It is then necessary to determine how the individual values will

be aggregated, so that they can be interpreted in sectoral terms. The results may be aggregated by the company type, but another way is to aggregate them according to the tasks performed by all of the companies. This way, the sector performance indicator for each task is calculated; the aggregation of these indicators will result in a sector performance indicator for each activity. By aggregating these, we can then calculate the performance indicators of the sub-processes. Finally, by aggregating the latter, we can establish the level of performance of each primary process. Regardless of the method, the representativeness of each company in the sector should be considered when aggregating the individual data. This representativeness can be determined by the company's turnover. Nevertheless, in both cases, the aggregation of the processes must be carried out in consideration of the importance of each process that is given by the stakeholders, who are participating in the AHP.

Once the ratings of the five main processes have been obtained and aggregated according to the weights defined by the AHP, the overall performance score of the SC can be achieved. Table 4 contains Kusrini et al.'s [64] proposal for rating supply chains' performance, according to a scale. The scale can be used to rate each company's performance and that of the sector, as well as the disaggregated results of the tasks, activities, processes, and sub-processes.

**Table 4.** Performance scale.

| Performance Values | Performance Indicator |
|---|---|
| <40 | Poor (P) |
| 40–50 | Marginal (M) |
| 50–70 | Average (A) |
| 70–90 | Good (G) |
| >90 | Excellent (E) |

Source: Kusrini et al. [64].

## 3. Case Study

We tested the proposed methodology in a case study of the Ecuadorian flower industry. Ecuador is currently the third-largest exporter of cut flowers worldwide. Although Ecuador had increased its exports up until 2019, it did so at a far lower rate in both value and volume than other flower-exporting countries. The subsequent fall became more marked in 2020, due to the restrictions brought on by the pandemic [65].

Flower production has, historically, been concentrated in the provinces of Pichincha, with 62% of the production, and Cotopaxi, with 21% of production. The rest of the country's provinces, including Guayas, Imbabura, and Azuay account for the remaining 17% [66]. Furthermore, it should be noted that the industry is presently in the midst of a wave of acquisitions. In the first quarter of 2021, the largest flower company in Ecuador (Hilsea Investments, with annual sales of around USD 50,000,000) was transferred to the investment company Sunshine Bouquet, which belongs to a group of the 500 largest companies in Colombia. Additionally, a number of other small firms, namely, Alma Roses, Sisapamba, Natuflor, Romaverde, Bellarosa, Rose Connection, Qualisa, and Florasani were taken over by the investment company Elite, one of the 500 largest companies in Ecuador [67].

For the case study, we selected a representative sample of floricultural companies from the Expoflores directory, where data was accessible. Specifically, the first 96 Ecuadorian flower companies (Order established according to the income data published by the Superintendencia de Compañías del Ecuador, https://www.supercias.gob.ec/portalscvs/, accessed on 30 April 2021) were chosen. According to the value of sales, these represented approximately 70% of the more than USD 800,000,000 turnover of the sector in 2019 [68]. As seen in Table 5, the turnover in these 96 companies varies from the largest to the smallest, i.e., from USD 12,000 to USD 47,000,000. The highest concentration of companies corresponds to those with a turnover of between USD 12,000 and USD 13,500,000. This group accounts for 93% of the total turnover in the industry.

**Table 5.** Frequency distribution by turnover (USD).

| Group | Lowest Turnover (USD) | Highest Turnover (USD) | Surveys Sent | Weight in the Whole Sector | Responses | Weight in the Sample | Participation Rate |
|---|---|---|---|---|---|---|---|
| 1 | 12,000 | 6,742,000 | 66 | 69% | 19 | 66% | 29% |
| 2 | 6,742,000 | 13,472,000 | 23 | 24% | 9 | 31% | 39% |
| 3 | 13,472,000 | 20,202,000 | 4 | 4% | | 0% | 0% |
| 4 | 20,202,000 | 26,932,000 | 2 | 2% | 1 | 3% | 50% |
| 5 | 26,932,000 | 33,662,000 | 0 | 0% | | 0% | |
| 6 | 33,662,000 | 40,392,000 | 0 | 0% | | 0% | |
| 7 | 40,392,000 | 47,122,000 | 1 | 1% | | 0% | 0% |
| | | | 96 | 100% | 29 | 100% | 100% |

Source: Authors' calculations.

Of the 96 companies to which we sent the survey, 29 answered. Table 5 shows the results of the frequency distribution analysis of the companies that answered the questionnaire. This analysis was performed to verify how representative they are. The frequencies were calculated according to the firms' turnovers. Most of the companies in the sample that answered were from the groups with the largest number of flower companies. Table 5 shows that the weight of the companies in the sample is similar to the weight of all companies in the Ecuadorian floriculture industrial sector in each turnover group.

We used a digital questionnaire (https://docs.google.com/forms/d/1GZDfiJLW5D7 IdsgrpjbXI696UlHAmH5tEOGQmmk-RKc/edit, accessed on 3 June 2020) to collect the preferences. Although various alternatives were available, we chose Google forms for this study. The form was sent to the companies' representatives by email, along with a letter explaining the study's purpose: to analyze the supply chain of Ecuador's flower industry, identify the key problems, and improve certain aspects.

The questionnaire was divided into four sections. The first section described the objective of the study and the survey and asked for the company's details. It also provided information on the Ecuadorian flower industry and the SC processes, as defined by the SCOR model. The following sections contained the questions about the supply chain processes. These were broken down to task levels. Respondents had to indicate which sub-processes, activities, and tasks they performed for each process.

Twenty-nine companies answered the survey, which accounted for approximately 20% of the total turnover of the selected sample, i.e., USD 180,000,000. Falcon Farms is the second-largest flower company, in terms of turnover, within this group of companies. The tasks were graded according to their fulfillment: a positive answer scored a one, and a negative answer scored a zero. Next, the average of the scores obtained for each sub-process was calculated, and then the average of the processes' scores was calculated for each of the 29 companies that answered the survey [51,69].

The importance of the SCOR model processes was determined by a group of stakeholders in the Ecuadorian floriculture sector by means of an AHP model. For this purpose, an online survey was undertaken. It was assumed that all members of the group had the same level of importance in the decision-making processes [70]. The stakeholders were: representatives of floriculture companies (6), supply chain teachers (2), experts in floriculture issues (1), and experts in quality control (1). The AHP methodology was applied to calculate the weights of the Level 1 metrics and the attributes of the SCOR model. A questionnaire was carried out that was divided into four sections. The first section described the study's objective and that of the questionnaire and requested information on the company or institution's identity. Furthermore, it included information on the Ecuadorian flower sector and descriptions of the performance attributes of the Ecuadorian supply chain, as well as the AHP hierarchy, with the objective of redesigning its elements, metrics, and processes, and an explanation of Saaty's scale for making the comparisons. The second section listed the questions related to the pairwise comparisons of the supply chain processes' attributes, in order to determine their importance (10 questions). The third section presented questions regarding the importance of the metrics for each attribute (7 questions). Finally, in the

fourth section, the questions about the relevance of the performance metrics to the supply chain processes were included (10 questions).

After collecting the preferences of the stakeholders by the processes considered, we aggregated the preferences of individuals and obtained the preferences matrix from stakeholders. This was used to calculate stakeholders' priorities. Table 6 shows the weights given by the stakeholders.

**Table 6.** Weighted results by process.

| No. | Process | Weight |
|---|---|---|
| 1 | Planning | 0.4051 |
| 2 | Procurement | 0.1986 |
| 3 | Manufacturing | 0.1735 |
| 4 | Distribution | 0.1381 |
| 5 | Return | 0.0847 |

Source: Authors' calculations.

We have a consistency ratio of $CR = 0.0209 \leq 0.10$, so the data comparing the main criteria pairs is appropriate and does not need to be re-evaluated.

Once the weights of the processes were calculated, the scores for each company were computed according to the results of the survey.

## 4. Results

We determined the supply chain performance level for each of the 29 companies that answered the questionnaire using the survey data. Then, the individual results were aggregated to determine the level of performance of the supply chain at the sector level. According to the analysis and the classification proposed by Kusrini [64], the 29 firms showed a good overall performance (see Tables 4 and 7). This rating was obtained because the score achieved by each of the five processes of the SCs of the companies studied was rated as "good" (G). However, it should be noted that, as the scores obtained for the processes were less than one, all procedures need to be reconfigured and improved.

**Table 7.** Calculation of the sector-level performance metrics.

| Process | AHP Weight | Average | Performance | Performance Metric |
|---|---|---|---|---|
| Planning | 0.4051 | 0.86 | G | 0.35 |
| Procurement | 0.1986 | 0.88 | G | 0.17 |
| Manufacturing | 0.1735 | 0.79 | G | 0.14 |
| Distribution | 0.1381 | 0.88 | G | 0.12 |
| Return | 0.0847 | 0.80 | G | 0.07 |
| | 1.00 | | | 0.85 |

Source: Authors' calculations.

The turnover of the 29 companies showed a relatively low correlation (0.08) with the SC performance index, which means that the supply chain performance does not explain the sales behavior.

When considering each of the SCOR processes at the sector level, it should be highlighted that the processes with the highest GAPs (GAP: gap or difference between the intended result and the actual result obtained by the research), weighted according to their weight, were planning (0.06) and manufacturing (0.04) (see Table 8).

**Table 8.** Supply chain performance GAPs at the sector level by process.

| Process | AHP Weight | Performance Metric | GAP |
|---|---|---|---|
| Planning | 0.4051 | 0.35 | 0.06 |
| Procurement | 0.1986 | 0.17 | 0.02 |
| Manufacturing | 0.1735 | 0.14 | 0.04 |
| Distribution | 0.1381 | 0.12 | 0.02 |
| Return | 0.0847 | 0.07 | 0.02 |
| | 1.00 | 0.85 | 0.15 |

Source: Authors' calculations.

To improve our analysis results and to better identify where the most critical points of the SC are, we also examined the sub-processes.

Regarding the analysis of the sub-processes, of the 16 sub-processes examined (shown in Figure 2), four were rated as "excellent" (E), eleven as "good" (G) and one as "average" (A) (see Table 9). Hence, the floriculture sector should pay attention to the sub-processes with "good" and "average" ratings.

**Table 9.** Supply chain performance GAPs at the sector level by sub-process.

| Code | Sub-Processes | AHP Weight | Performance Metric | Performance | GAP |
|---|---|---|---|---|---|
| **1** | **Planning process (PLAN)** | 41% | | | |
| 1.1 | Supply chain planning | | 80% | G | 20% |
| 1.2 | Linearity of the supply chain (alignment of supply and demand) | | 85% | G | 15% |
| 1.3 | Inventory management | | 92% | E | 8% |
| **2** | **Procurement process (SOURCE)** | 20% | | | |
| 2.1 | Strategic sourcing | | 92% | E | 8% |
| 2.2 | Supplier management | | 89% | G | 11% |
| 2.3 | Buying products and services | | 82% | G | 18% |
| 2.4 | Management of inbound logistics | | 89% | G | 11% |
| **3** | **Manufacturing process (MAKE)** | 17% | | | |
| 3.1 | Supplier relationships and collaboration | | 72% | G | 28% |
| 3.2 | Product | | 92% | G | 8% |
| 3.3 | Development of the supply chain infrastructure | | 74% | G | 26% |
| 3.4 | Sales logistics | | 77% | G | 23% |
| **4** | **Distribution process (DELIVER)** | 14% | | | |
| 4.1 | Storage and compliance | | 90% | E | 10% |
| 4.2 | Customer and business partner management | | 86% | G | 14% |
| **5** | **Return process** | 8% | | | |
| 5.1 | Receiving returned goods and storage | | 78% | G | 22% |
| 5.2 | Repair and refurbishment | | 93% | E | 7% |
| 5.3 | Customer expectation management | | 69% | A | 31% |

Source: Authors' calculations.

By evaluating the different activities of each sub-process, we assigned each activity the corresponding Kusrini rating. As a result, several activities with "good," "average,"

and "marginal" ratings need to be improved. The most critical activities, which received the lowest ratings, are:

(1)   Activities with a "marginal" rating:

-   One-to-one (task) training, i.e., there is a training program for new employees (45%).

(2)   Activities with an "average" rating:

-   Methods for estimating needs related to the task, i.e., statistical techniques are used to estimate the needs and validate the data sources employed to make these estimates (59%);
-   The authorization of casual purchases related to each task, i.e., casual purchases that do not exceed a certain amount, as defined by the company, are authorized (66%);
-   Feedback from customers concerning each task, i.e., the company undertakes customer satisfaction surveys at least once a year (52%);
-   Workforce and skill versatility, i.e., workers regularly switch jobs since they know how to do them (66%);
-   Sales management related to each task, i.e., the company undertakes customer satisfaction surveys (62%);
-   Returned goods management, i.e., there is a system for classifying returned goods (69%);
-   Accounting transactions, i.e., inventory adjustments are regularly carried out as part of the returned goods process (69%).

By examining the results at the company level, we can determine which companies in the sector are having the greatest problems and, therefore, need to optimize their processes. It also enables us to see which processes in each company are performing poorly. The analysis at the company level can be done individually, or by groups of companies. Table 10 shows that no company received a "poor" or "marginal" rating; four companies were rated with "average" performances, thirteen companies gave a "good" performance, and twelve gave an "excellent" performance.

**Table 10.** Summary of the SC performance metrics for the 29 companies that answered the survey.

| Performance Values | Performance Indicator | No. of Companies | Performance Metric |
|---|---|---|---|
| <40 | Poor | 0 | |
| 40–50 | Marginal | 0 | |
| 50–70 | Average | 4 | 0.60 |
| 70–90 | Good | 13 | 0.84 |
| >90 | Excellent | 12 | 0.94 |
| | Total | 29 | 0.85 |

Source: Authors' calculations.

Together, the four flower companies with an "average" performance rating (see Table 10) achieved a performance level of 60%. Their turnover ranged from USD 12,374 to USD 2,400,000 during 2012–2019. The process with the highest GAP was planning in the four companies, at 0.19; the remaining processes showed similar GAPs, close to 0.05 (see Table 11).

**Table 11.** Supply chain performance GAPs of groups of companies that answered the survey by index performance.

| Process | AHP Weight | 4 Companies between 50 and 70 | | 13 Companies between 70 and 90 | | 12 Companies Higher Than 90 | |
|---|---|---|---|---|---|---|---|
| | | Performance Metric | GAP | Performance Metric | GAP | Performance Metric | GAP |
| Planning | 0.4051 | 0.21 | 0.19 | 0.35 | 0.05 | 0.39 | 0.01 |
| Procurement | 0.1986 | 0.14 | 0.06 | 0.18 | 0.02 | 0.18 | 0.01 |
| Manufacturing | 0.1735 | 0.12 | 0.06 | 0.13 | 0.04 | 0.15 | 0.03 |
| Distribution | 0.1381 | 0.10 | 0.04 | 0.11 | 0.02 | 0.14 | 0.00 |
| Return | 0.0847 | 0.04 | 0.05 | 0.06 | 0.02 | 0.08 | 0.00 |
| | 1.00 | 0.60 | 0.40 | 0.84 | 0.16 | 0.94 | 0.06 |

Source: Authors' calculations.

The thirteen floriculture companies that achieved a "good" performance rating have a turnover ranging from USD 118,000 to USD 26,400,000 during 2012-2019. All together, they achieved a performance of 84%. Planning and manufacturing stand out in these companies as the processes with the highest GAPs (see Table 11).

The minimum turnover of the twelve flower companies that achieved an "excellent" rating was USD 636,000, and the maximum turnover was USD 9,500,000 in the 2012–2019 period. Together, the companies achieved a performance level of 94% (see Table 10), which can be considered as "excellent."

The process that had the most GAPs was the manufacturing process, whereas the distribution and return processes did not show any GAPs (see Table 11).

Regarding the sub-processes' performance, in the group of companies (4) with average performances, two sub-processes obtained a "poor" rating, eight sub-processes obtained an "average" rating, and six sub-processes obtained a "good" rating. Therefore, no sub-processes achieved an "excellent" rating in this group.

The sub-processes carried out by the group of companies with the lowest scores (i.e., "poor") were supply chain planning (38%) and customer expectation management (25%). The following sub-processes received an "average" rating: the linearity of the supply chain (the alignment of supply and demand) (50%), inventory management (69%), strategic sourcing (65%), buying products and services (63%), the development of the supply chain infrastructure (63%), sales logistics (58%), receiving returned goods and storage (50%), and repair and refurbishment (50%). Those considered to have a "good" performance were supplier management (75%), the management of inbound logistics (70%), supplier relationships and collaboration (75%), the product (75%), storage and compliance (70%), and customer and business partner management (75%).

In the group of the companies (13) that achieved a good performance level, the analysis by sub-processes resulted in five sub-processes with an "excellent" rating, eight with a "good" one, and three with an "average" rating.

In this group, the sub-processes with an "excellent" rating were inventory management (96%), strategic sourcing (95%), supplier management (90%), the product (92%), and repair and refurbishment (100%). Those with a "good" rating were supply chain planning (81%), the linearity of the supply chain (the alignment of supply and demand) (83%), buying products and services (85%), the management of inbound logistics (88%), the development of the supply chain infrastructure (73%), sales logistics (74%), storage and compliance (89%), and customer and business partner management (77%), while those with average scores were supplier relationships and collaboration (69%), receiving returned goods and storage (65%), and customer expectation management (62%).

In relation to the group of companies rated as "excellent", the analysis of sub-processes resulted in twelve sub-processes with an "excellent" rating, and four with a "good" rating.

Regarding the third SCOR level, of the 48 activities studied, 24 achieved an "excellent" rating, 16 activities showed a good performance level, seven activities exhibited an average level, and one activity was considered to have a "marginal" performance level. Thus, the

activities that the floriculture sector should pay more attention to are those with "good, average, and marginal" ratings, which accounted for 51% of the studied activities.

## 5. Discussion

Our proposed methodology shows that it is possible to analyze the performance of the supply chain at the sectoral level by applying the SCOR model and the AHP in a representative sample of companies in the sector. In previous research, these analyses were more limited. The majority did not disaggregate the SCOR model, and only studied the first level, regarding the processes [44,46–49]. Other studies were on unique companies and the results cannot be viewed as sectoral results [42,46,47]. There are some studies where the proposed methodology only studied a stage in the supply chain, and only one element in this stage. For example, Wang et al. [43] applied the model to a raw material supplier. Other works analyzed the sector and does not use company data. These used focus groups or stakeholders' opinions instead [46]. Sutoni et al. [47] used observations, interviews, literature reviews, and information or dates, but these were from a single company.

In general, an analysis of the individual processes, sub-process, activities, and tasks would enable us to identify which process, sub-processes, activities, or tasks, are more critical why they are more critical, what the causes are, and hence, how they can be corrected. The methodology proposed makes possible this analysis at the individual level, for each company, and at the sectoral level.

The proposed SC performance analysis method can be used with any company and with any industry, since it allows the evaluation of groups of companies that make up an industry or represent it, by aggregating the individual values. Additionally, it is a tool that determines where problems lie and their causes. It also helps to increase the competitiveness of firms and industries, and achieves long-term goals by supporting company managers, governments, policymakers, and every industry in the design of policies and measures to fix issues. Managers can use the results to benchmark their company's competitiveness and performance against other companies in their sector, or in sectors with similar characteristics. In the policy field, sector-level analyses can be used for planning purposes.

## 6. Conclusions

This study contributes to the current literature with a methodological proposal that uses the SCOR model, combined with an analytical hierarchy process (AHP) to measure the performance of supply chains within a given sector. We applied this methodology to individual flower companies to assess the degree of compliance of their supply chain (SC) processes and activities, with the standards set by the SCOR model regarding SC performance. In addition, we determined which tasks or activities in each company were not carried out and traced the origin of potential problems back to specific SC sub-processes, which should be checked. Moreover, the aggregation of performance data at the individual level enabled us to assess the performance at the sector level.

Here, we employed the proposed methodology to identify, calculate, and handle potential SC performance issues in the Ecuadorian floriculture industry. By conducting an in-depth study of Ecuadorian flower companies, we have been able to draw a comprehensive picture of this industry.

Based on the results for the 29 companies that answered the survey, the SC performance of the Ecuadorian flower sector is 85%. The results showed that all processes need to be improved, especially the planning and manufacturing processes. When analyzing the flower companies by groups according to their rating, the planning, procurement, and manufacturing processes with an "average" rating (50–70) showed large performance GAPs. Meanwhile, the planning and manufacturing processes of companies with a score of 70–90, which is considered "good", had the largest performance GAPs. Moreover, within the group of companies with a performance score that was higher than 90, the manufacturing process is the most critical.

Therefore, Ecuadorian flower companies should work on the first five SCOR processes, applying the standards suggested in the model. To excel, they should work on all processes, which also depend on external factors, in order to improve the flower industry's supply chain.

When conducting studies such as this one, the sample must be as representative of the industry as possible. Therefore, in general, obtaining a high response rate allows for a better analysis and results that reflect the realities of the sector. Hence, the scope of future studies about the Ecuadorian flower industry must be expanded to include a larger number of companies and a broader field of analysis, considering performance attributes such as reliability in compliance, the speed of responses, agility, costs, and the efficient management of assets and their components.

**Author Contributions:** Conceptualization, I.M.-P. and I.G.-P.; methodology, I.M.-P.; software, L.O.R.M.; validation, I.M.-P. and I.G.-P.; formal analysis, L.O.R.M.; investigation, L.O.R.M.; resources, L.O.R.M.; data curation, L.O.R.M.; writing—original draft preparation, L.O.R.M.; writing—review and editing, I.M.-P. and I.G.-P.; visualization, I.G.-P.; supervision, I.M.-P.; project administration, I.M.-P. All authors have read and agreed to the published version of the manuscript.

**Funding:** This research received no external funding.

**Institutional Review Board Statement:** Not applicable.

**Informed Consent Statement:** Not applicable.

**Data Availability Statement:** I Superintendencia de Compañías del Ecuador (https://www.supercias.gob.ec/portalscvs/ accessed on 30 April 2021).

**Conflicts of Interest:** The authors declare no conflict of interest.

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
