# Peer review of "Measuring the Supply Chain Performance of the Floricultural Sector Using the SCOR Model and a Multicriteria Decision-Making Method"

_horticulturae, doi:10.3390/horticulturae8020168_

Round 1
Reviewer 1 Report
The topic of the paper is quite interesting and up to date. It is generally well and clearly written. Some points however needs more elaboration.
Some detailed remarks:
In the introduction: In general, as I understand, the main scientific problem of the paper is methodology based. Therefore I suggest to start introduction with explaining the scientific problem and the knowledge gap (as the method is concerned) and how the paper will contribute to this knowledge gap. The illustration of this measurement on the flower chain is a second importance and should be later (after line 181) – same as Equador’s description, I supposed chosen as the good case of the flower chain and illustration of the method.
I miss also a bit contribution to the theory – why this approach is better and how refers to the theory of SC efficiency measurement.
53-56: does the authors mean that the chains developed last 40 years, only? and especially after pandemic! Why, how?
In the methodology section – the figure 1 has moderate quality and graphically is also not well prepared.
Figure 2 should be limited only to the first column (1.1-5.X) and 1.x.x – 5.x.x.) and the remaining part (detailed questions) should be provided in annex in well edited MS Word table. Practically it might be combined together with table 3. Besides there is no clear explanation why in table 3 and in text in line 240 there are 4 processes and in figure 2 – 5 processes. It misleads the reader
296: case study? Or casen study?
313: how representativeness has been measured – for which criterion is the sample representative? 70% of turnover represents only the biggest companies – so the sample is not representative for remaining 30% usually small holdings. In my opinion authors shouldn’t call the sample representative unless providing the proof for the criterion of statistical representativeness.
Table 6. What was the criterion to select 1-7 groups by turnover – why such values are selected for the group distinction.
347 – how many stakeholders participated, what were they qualification to judge the weights.
There is no discussion section as required by the journal, where you may discuss the methods’ s testing results with the other papers mentioned in the introduction, using these and similar methods, showing advantages of your approach, I insist in adding it.
In conclusion part should not be any references and detailed results as they are now.
Author Response
Dear reviewer,
I am sending Horticulture our manuscript entitled " MEASURING SUPPLY CHAIN PERFORMANCE OF THE FLORICULTURAL SECTOR USING THE SCOR MODEL AND A MULTICRITERIA DECISION-MAKING METHOD ".
We thank you for their comments and suggestions, which we understand will improve the manuscript quality, and for this reason, we have taken them into account. We have made the revisions for our paper, and we send you now, the manuscript to consider your comments. We did not want to refute any point from comments. We attached a list of changes which was been raised in our revised manuscript.
Thank you very much for your care and attention.
We wait for a decision on our manuscript as soon as possible.
Sincerely yours

Reviewer 2 Report
The proposed manuscript aims to highlight the usefulness of studying the 10 level of performance of supply chains (SC) through combination of a disaggregated SCOR (Supply Chain Operations Refer-12 ence) model with a multicriteria decision-making approach (AHP). More in particular, the study analysed the case study the Ecuadorian floriculture industry.
Some suggestions seem to be necessary to improve the paper. A minor revision is required.
The paper is well written and easy to read.
first of all, in the introductory part, other papers that have analyzed the performance of the more integrated and collaborative SC, also in innovation processes, should be considered. Among these manuscripts, I suggest:
- Karantininis, K.; Sauer, J.; Furtan, W.H. Innovation and integration in the agri-food industry. Food Policy
2010, 35, 112–120.
- Triguero, A.; Fernández, S.; Sáez-Martinez, F.J. Inbound open innovative strategies and eco-innovation in the
Spanish food and beverage industry. Sustain. Prod. Consum. 2018, 15, 49–64.
- Stanco, M., Nazzaro, C., Lerro, M., & Marotta, G. (2020). Sustainable Collective Innovation in the Agri-Food Value Chain: The Case of the “Aureo” Wheat Supply Chain. Sustainability, 12(14), 5642.
- Zilberman, D.; Lu, L.; Reardon, T. Innovation-induced food supply chain design. Food Policy 2019, 83,
289–297.
The numbering is missing from the "Result" paragraph.
Author Response

(The authors gave the same response as above.)

Reviewer 3 Report
I think that the work is interesting, that it contributes to the literature in the field. The paper is well organized and deals with an important research topic, but, some minor improvements could to be made.
My suggestions:
- I don't think that Table 3 and Figure 2 are solved in the best way. Table 3 and Figure 2 should be combined either as an unique table or as an unique figure (as the authors decide) that will include both processes and sub-processes and activities and tasks. Then correct the numbering of figures / tables.
- I think it would be clearer if Tables 12, 13 and 14 are combined into one table that will contain all the data entered in Tables 12, 13, 14. After that, adjust the comments to this change.
Author Response

(The authors gave the same response as above.)

Reviewer 4 Report
1.The first part of the author is a little lengthy. It is suggested that the authors divide it into two parts: introduction and literature review.
- What are the real pain points of the problem being studied?Where is the research gap? It is suggested that the author condense further.
- The authors make a detailed table about the disaggregationof the SC processes and sub-processes based on the SCOR model. But how does the indicators of the sub-processes and activities come? If they all come from López[63], what is the author's theoretical contribution?
Author Response
Dear reviewer,
I am sending Horticulture our manuscript entitled " MEASURING SUPPLY CHAIN PERFORMANCE OF THE FLORICULTURAL SECTOR USING THE SCOR MODEL AND A MULTICRITERIA DECISION-MAKING METHOD ".
We thank you for their comments and suggestions, which we understand will improve the manuscript quality, and for this reason, we have taken them into account. We have made the revisions for our paper, and we send you now, the manuscript to consider your comments. We attached a list of changes which was been raised in our revised manuscript.
Thank you very much for your care and attention.
We wait for a decision on our manuscript as soon as possible.
Sincerely yours

Round 2
Reviewer 4 Report
- P1, line11, “SCOR (Supply Chain Operations Reference)” should be corrected as “Supply Chain Operations Reference (SCOR)”. When defined for the first time,the acronym/abbreviation/initialism should be added in parentheses after the written-out form.
- P1, line23, “developing” or “implementing” should be added between “The study’s main contribution is” and “a general framework”.
- The abstract should be a total of about 200 words maximum. The abstract section does not adequately state the research background, and the necessity and importance of the research is not shown enough. The author needs to place the question addressed in a broad context and highlight the purpose of the study.
- The purpose and significance of this study should be highlighted again in the end of the introduction section.The author should pay attention to the length of the introduction section. This section needs to be shortened via deleting some sentences and putting parts of model introduction in the next section.
- In the section of the case study, of the 96 companies to which the author sent the survey, only 29 answered. There is a certain possibility that only companies with high level of supply chains performance answered the questionnaire. The author attempted to verify the representative of the sample, however, only selecting “turnover” as the indicator is not persuasive.
- The author disaggregated the supply chain processes into several sub processes, however, in the section of the results, there is no AHP weight of sub factors. The analysis process of AHP is too simple. In addition, there is no consistency test of AHP.
- The results and the discussion are suggested to bedescribed separately.
Author Response
Dear reviewer,
I am sending our manuscript entitled.
We thank you for their comments and suggestions, which we understand will improve the manuscript quality, and for this reason we have taken them into account. We have made the revisions for our paper, and we send you now, the manuscript considered your comments. We attached a list of refusals and changes which was been raised in our revised manuscript.
Thank you very much for your care and attention.
We wait for a decision on our manuscript as soon as possible.
Sincerely yours
ANSWERS TO REVIEWERS’ COMMENTS
|
LIST OF CHANGES OR REBUTTAL |
|
|
Reviewer #4: |
|
1 |
P1, line11, “SCOR (Supply Chain Operations Reference)” should be corrected as “Supply Chain Operations Reference (SCOR)”. When defined for the first time,the acronym/abbreviation/initialism should be added in parentheses after the written-out form. |
Following your instructions, we have corrected the line 11 |
2 |
P1, line23, “developing” or “implementing” should be added between “The study’s main contribution is” and “a general framework”. |
Following your instructions, we have added developing between “The study’s main contribution is” and “a general framework”. |
3 |
The abstract should be a total of about 200 words maximum. The abstract section does not adequately state the research background, and the necessity and importance of the research is not shown enough. The author needs to place the question addressed in a broad context and highlight the purpose of the study. |
Following your instructions, we have made the abstract shorter (199 words) We include in the abstract, in this order: Aim of study/Area of study/Material and methods/Main results/ Research highlights: |
|
|
This study aims to highlight the usefulness of studying the level of performance of supply chains (SC) at the sectoral level in greater detail through the combination of a disaggregated Supply Chain Operations Reference (SCOR) model with a multicriteria decision-making approach, concretely AHP to adjust the analysis to the particularities of the sector under study. The methodology was applied to the Ecuadorian flower industry, and the data for the analysis was from a survey of a group of companies that represent this sector. In addition, a focus group of SC experts weighted the model constructs as part of the Analytic Hierarchy Process (AHP), and then the performance level for each construct was determined. According to the results, methodologies allow classified companies by their performance and estimate the performance of the aggregate sector. The processes that Ecuadorian flower companies need to improve are planning, procurement, and manufacturing. The study’s main contribution is developing a general framework for measuring the overall performance of SCs and how the results are obtained. This tool could help managers, consultants, industries, and governments to assess the performance of SCs, as well as improve SC management in order to increase the sector’s competitiveness in the international market. |
4 |
The purpose and significance of this study should be highlighted again in the end of the introduction section. |
Following your instructions, we have highlighted the significance of this study at the end of the introduction section |
|
The author should pay attention to the length of the introduction section. This section needs to be shortened via deleting some sentences and putting parts of model introduction in the next section. |
Following your instructions, we have shortened the introduction section. We have moved parts of the description of the model in the introduction to the methodology |
5 |
In the section of the case study, of the 96 companies to which the author sent the survey, only 29 answered. There is a certain possibility that only companies with high level of supply chains performance answered the questionnaire. The author attempted to verify the representative of the sample, however, only selecting “turnover” as the indicator is not persuasive. |
Based on the characteristics data of the companies, we had considered that turnover was a good data for classifying by size. |
6 |
The author disaggregated the supply chain processes into several sub processes, however, in the section of the results, there is no AHP weight of sub factors.
|
The disaggregation of the processes into sub-processes, activities, and tasks, results, as can be seen in Figure 2, in many elements in the hierarchy that would be difficult to assess through the AHP methodology. On the other hand, by giving weight to the processes, the sub-processes, activities, and tasks that derive from each of them are also being weighted. In subsequent studies, the weighting of the disaggregation of the processes could be analyzed. |
|
The analysis process of AHP is too simple. In addition, there is no consistency test of AHP. |
Following your instructions, we have added the results of the consistency analysis of the AHP |
7 |
The results and the discussion are suggested to be described separately. |
Following your instructions, we have separated results and discussion |
We wait for a decision on our manuscript as soon as possible.
Sincerely yours
ANSWERS TO REVIEWERS’ COMMENTS

Round 3
Reviewer 4 Report
Thanks for the efforts made by the author to improve the quality of the article.